# A new framework based on features modeling and ensemble learning to predict query performance

**Mohamed Zaghloul** \*, **Mofreh Salem, Amr Ali-Eldin**

Computer Engineering and Control Systems Dept, Faculty of Engineering Mansoura, Mansoura, Egypt

\* mmustafa@nxn.ae

**Data Availability Statement:** Supporting Information files.

**Funding:** The authors received no specific funding for this work.

## Abstract

A query optimizer attempts to predict a performance metric based on the amount of time elapsed. Theoretically, this would necessitate the creation of a significant overhead on the core engine to provide the necessary query optimizing statistics. Machine learning is increasingly being used to improve query performance by incorporating regression models. To predict the response time for a query, most query performance approaches rely on DBMS optimizing statistics and the cost estimation of each operator in the query execution plan, which also focuses on resource utilization (CPU, I/O). Modeling query features is thus a critical step in developing a robust query performance prediction model. In this paper, we propose a new framework based on query feature modeling and ensemble learning to predict query performance and use this framework as a query performance predictor simulator to optimize the query features that influence query performance. In query feature modeling, we propose five dimensions used to model query features. The query features dimensions are syntax, hardware, software, data architecture, and historical performance logs. These features will be based on developing training datasets for the performance prediction model that employs the ensemble learning model. As a result, ensemble learning leverages the query performance prediction problem to deal with missing values. Handling overfitting via regularization. The section on experimental work will go over how to use the proposed framework in experimental work. The training dataset in this paper is made up of performance data logs from various real-world environments. The outcomes were compared to show the difference between the actual and expected performance of the proposed prediction model. Empirical work shows the effectiveness of the proposed approach compared to related work.

## Introduction

A query optimizer attempts to make a comparable performance estimate. It attempts to forecast the time spent on a performance metric. It necessitates access to the most recently processed performance statistics. In theory, this requirement creates a significant overhead for supplying the necessary statistics to the core engine. Many factors influence query

**Competing interests:** The authors have declared that no competing interests exist.

performance, such as data stores, data architecture, environmental specifications, and query syntax [1]. Machine learning is becoming more popular in database improvement. So, which machine learning technique will be responsible for query performance prediction? All of this will be covered in this paper.

## Data store

Data can be stored in the Hadoop file system (HDFS) or Google file system (GFS) or in a database such as a relational or a columnar database [1, 2]. The basic identifier for distinguishing these databases is that the columnar database stores each column in a sequence so that all the values of these columns are adjacent to each other. A relational database, on the other hand, stores the data as a row in an adjacent manner, with the row's various column values parallel to each other. Columnar databases are best suited for analytical workloads, whereas relational databases are better suited for transactional workloads. Real-time analytics is enabled by in-memory columnar databases. Incorta [3], for example, is faster than Oracle database management system (DBMS) [3].

## Data architecture

In addition to data stores having an impact on query performance, data architecture is also a factor. The new data architecture is a hybrid of the Lambda and Kappa architectures [4, 5]. In summary, the Lambda Architecture is used in various layers for batch and stream processing [6, 7]. The batch layer of the Lambda Architecture manages historical data with fault-tolerant distributed storage, ensuring a low possibility of errors even if the system crashes. It has a good balance of speed and reliability, as well as a fault-tolerant and scalable data processing architecture. The Lambda Architecture has the disadvantage of resulting in coding overhead due to the involvement of comprehensive processing. It is difficult to migrate or reorganize data that has been modelled using the Lambda Architecture. The Kappa Architecture, on the other hand, is regarded as a more straightforward alternative to the Lambda Architecture because it employs the same technology stack to handle both real-time stream processing and historical batch processing. Both architectures require the storage of historical data to enable large-scale analytics. They are also useful for addressing "human fault tolerance," in which problems with the processing code (either bugs or simply known limitations) can be overcome by updating the code and running it again on the historical data. The main distinction with the Kappa Architecture is that all data is treated as a stream; thus, the stream processing engine serves as the sole data engineering engine. The lack of a batch layer in the Kappa Architecture may result in errors during data processing or database updates, necessitating the use of an exception manager to reprocess the data or perform reconciliation [7–9].

## Query performance prediction is regression problem

The prediction of query performance is regarded as a regression problem. Different regression algorithms, such as linear regression, lasso regression, neural networks, ridge regression, and so on, can be considered here [10]. Before applying scoring to the model, the initial model development approach is to have a training dataset and a cross-validation dataset. Simple linear regression, polynomial regression, support vector regressor, ridge regression, and lasso regression are some regression algorithms that can be used to solve such problems. There must be a linear relationship between the independent and dependent variables, no multicollinearity within the independent variables, and the residual mean must be zero [11, 12]. Polynomial regression converts the original features into polynomial features of a specific degree before applying linear regression. With a few minor exceptions, support vector regression

follows the same principles as support vector machine for regression [13]. In the case of regression, a tolerance epsilon margin is a fixed approximation to the SVM. The main idea, however, is to minimize errors by individualizing the hyperplane, which maximizes the margin. There are also linear and non-linear SVR [14]. When the data is multicollinearity, ride regression is used (independent variables are highly correlated). Even though the ordinary least squares (OLS) estimates are unbiased in multicollinearity, their variance is large, causing the observed value to deviate significantly from the true value. Ridge regression reduces errors by adding a degree of bias to the regression estimates. L2 has a tendency to evenly shrink coefficients [15]. When you have collinear/co-dependent features, this is useful. The loss function equation with regularization is as follows:

$$Loss\ function = RSS(W) + \lambda ||\ w\ || \tag{1}$$

We have a set of metrics for measuring things. The mean square error (MSE) [16] is the first metric. It is simply the average of the squared difference between the target value and the regression model's predicted value. Because it squares the differences, it penalizes even minor errors, leading to an overestimation of how bad the model is. It is preferred over other metrics because it is different and thus can be optimized more effectively. The MSE is calculated as follows:

$$MSE = \frac{1}{n \sum_{i=1}^{n} (\mathrm{y} - \bar{\mathrm{y}})^2} \tag{2}$$

The root mean squared error (RMSE), the most used metric for regression tasks, is the square root of the average squared difference between the target value and the value predicted by the model [16]. It is preferred in some cases because errors are squared before averaging, imposing a high penalty on large errors. This means that the RMSE can be useful when large errors are undesirable. The RMSE is calculated as follows:

$$RMSE = \sqrt{1/N \sum_{i=1}^{n} (y_i - \widehat{y_i})^2} \tag{3}$$

The third metric is the mean absolute error (MAE), which is the absolute difference between the target value and the value predicted by the model [16, 17]. The MAE is more resistant to outliers and does not penalize errors as harshly as the MSE. The MAE is a linear score, which means that all individual differences are equally weighted. It is not appropriate for applications where you want to pay more attention to outliers. The MAE equation is as follows:

$$MAE = 1/N \sum_{i=1}^{n} |y_i - \widehat{y_i}| \tag{4}$$

## Ensemble learning

Ensemble learning techniques are also being considered for use in the solution of regression problems. Ensemble learning entails combining multiple models and can also be used to classify and predict business problems. Ensemble learning is based on a bagging or boosting approach. In bagging, all the individual models are built in parallel. Each individual model is distinct from the others. All observations in all samples will be treated equally in this approach. Bagging aids in the reduction of overfitting. Using the majority voting method, we average all the model's outputs. The predicted value for regression models will not be optimized. If any of the models deviates more than the others, the output value will be the average of all the models. Random forests are a type of bagging algorithm. All the individual models in the boosting ensemble approach are built sequentially, which means that the outcome of the first model is passed on to the next model, and so on. Boosting is an ensemble-based learning algorithm that

converts weak learners to strong estimators by training ML models sequentially one after the other, with each iteration attempting to correct the model's errors from the previous iteration. In boosting, on the other hand, once the first model is built, we know the errors of that model. So, when we pass this first model to the next model, the goal is to reduce errors even more. Boosting reduces bias because each model in the sequential chain attempts to reduce the errors of the previous model. To reduce the errors in each sequential model, we can use multiple loss functions. The boosting method is prone to overfitting. As the models are built sequentially, all of them attempt to reduce training errors [18–20].

This paper aims to propose a query performance prediction framework that can be used to estimate query performance. The proposed framework can then be used as a query performance simulator to improve query syntax, optimize the allocated environment resources, and enhance the data architecture features used to execute the query. The main contribution is to model all the query features that influence query execution, feeding the ensemble learning technique with all the query-impactful features, to develop a query performance prediction model. The proposed framework can then be used as a query performance simulator to improve query syntax, optimize the allocated environment resources, and enhance the data architecture features used to execute the query. Section two discusses the related background, research motivation, and explains the challenges encountered while using the traditional approach, and sections three, four, and five introduce the proposed approach, experiment work, results, and discussion. Finally, section six summarizes this work.

## Related work

A query optimizer is one that attempts to make a similar estimate of performance. It tries to predict a performance metric based on the amount of time elapsed. It necessitates access to runtime performance statistics, the vast majority of which are processed. The need for the core engine to provide the necessary statistics creates a significant overhead in theory. Machine learning is increasingly being used to improve databases by incorporating regression models. Regression models strive for overall prediction accuracy, which means that the model expects data to be used in the same way that it was used for training. In data management, it is argued that query distributions do not always obey data characteristics. Queries that use selection operators explore data subspaces, eliminating the need to build a highly detailed model of the entire data collection. Given the increasing demand for regression models to be integrated with database management systems (DBMS) [21–23], regression models optimize a loss function, whereas the query-centric approach seeks to reduce the error rate for a typical single query in a planned workload. Under this assumption, the query data is the same as the training data. In general, query workloads have been presented in data systems research. This section will present a number of related works that demonstrate various approaches and frameworks proposed for the SQL query performance prediction problem.

The first approach emphasized the difficulty of selecting the best regression model to provide the best prediction result for a specific query because the model's behavior varies depending on the algorithm, mathematical model, and dataset. So, a question is first transferred to a classifier, which is pre-trained. Since the classifier "knows" the prediction capacity of each model in the different data spaces, the query will be allocated to the model that performs better in the query's data space. Unlike traditional data-mining approaches, only one machine learning model is invoked for each query [23, 24]. To conclude, for different data sets, different regression models are better performing and, more interestingly, for different data subspaces within them. This applies to simpler models and, perhaps unexpectedly, to advanced regression models of the ensemble. With minimal overheads, accuracy improvements are achieved,

even with huge data sizes. The same can be concluded from this approach. It combines various techniques to address each step of the problem of predictive estimation time [25–27]. This approach uses a few features to quickly predict the response time for a query with an acceptable error, according to the analysis of the obtained results. To improve the approach, this approach focuses solely on resource utilization (CPU, I/O); thus, the model features generalization with different environment specifications, query syntax complexities, and other factors that influence query performance [26, 27].

An alternative approach is based on performing feature selection to reduce the dimensionality of the input vectors. Step (A), extracting feature vectors from a mass of SQL data. The feature extraction is based on a bag of words (BoW) technique. Step (B) The model selection process receives such a dataset, (1) adjusts the parameters and performs a grid search for the most appropriate specific parameters. (2) each trained model is then evaluated using the K-fold cross validation (cv) to estimate the associated error. (C) The specified error metric for the K-fold CV method is the mean absolute error between the expected value and the projected target values for each partition. (D) Therefore, by choosing a subset of features that best characterize the phenomenon, a method of dimensional reduction is applied. The selection function uses models with tuned parameters derived from the selection model. This research presented a machine learning approach to query response time estimation in the cloud. It incorporates various techniques to address each stage of the problem of predictive time estimation. This work was analyzed in the sense of the cloud world. Based on the interpretation of the results obtained, the method uses a few features to easily predict the query response time with limited features [28–30].

Another research approach to using kernel canonical correlation analysis (KCCA) replaces Euclidean dot products with kernel functions [31–33]. Kernel functions are at the heart of many recent developments in machine learning, as they provide expressive, computationally tractable notions of similarity [31–33]. The KCCA model approach is based on the kernel function to compute a "distance metric" between every pair of query vectors and performance vectors. In this work, the researchers made use of the commonly used Gaussian kernel. It describes the steps involved in using KCCA to create a predictive model. The first step is to create feature vectors for all the points in the two datasets and then correlate them. For query prediction, the researcher builds a vector capturing query features and a vector capturing performance characteristics for each query in the training data set. The researcher combines these vectors into a query feature matrix with one row per query vector and a performance feature matrix with one row per performance vector. It is important that the corresponding rows in each matrix describe the same query. KCCA then uses a kernel function to compute a "distance metric" between every pair of query vectors and performance vectors. Thus, intuitively, KCCA finds correlated pairs of clusters in the query vector space and the performance vector space. The prediction is done in three steps. First, we create a query feature vector and use the model to find its coordinates on the query projection. The researcher then infers its coordinates on the performance projection and uses the k nearest neighbors in the query projection to do so. Finally, the map takes place from the performance projection back to the metrics when we want to predict. Finding a reverse mapping from the feature space back to the input space is known as a hard problem, because of the complexity of the mapping algorithm and because of the dimensionality of the feature space that can be much higher or lower than the input space [33, 34].

The challenges of previous approaches, which relied solely on hardware specifications features such as (CPU, I/O, allocated RAM) to develop query performance prediction models, were highlighted in previous related work. The previously proposed models are not generalized enough to be used as a performance simulator to enhance query features that influence

performance. Further, proposed hybrid models for determining which regression algorithm will fit into the distributed data space and produce the lowest error can be seen as a work-around. However, they are not optimal. As a consequence, we need to employ a more sophisticated regression algorithm.

## Proposed framework

Fig 1 depicts the framework flow and explains the purpose of each step in the flow. There are two flows in the proposed framework: the training flow, which focuses on modeling the query features and then training the machine learning model, and the score flow, which explains how to use a trained model in any other environment with a new SQL statement syntax. The diagram is divided into layers, including the metadata source layer, the feature modeling layer, the training/ scoring layer, and the consuming layer. The metadata source layer is used to list the query dimensions that are being considered for extracting the query features. Step (1) is used to model the SQL syntax features in the modeling features layer. Step 2 brings together all the other features that will be used to train the predictive model. To ensure that all input features are numbers, step 3 employs engineering features such as encoding categorical features into a number, reformatting a number, and deleting redundant features. Step (4) employs the XGBoost algorithm to train the model. Step (5) makes use of the built-in feature's importance in the XGBoost algorithm. The mean absolute error, root-mean-square error (RMSE), and average percentage difference between actual and predicted performance will be considered as regression performance metrics. Steps 6 through 9 of the score flow will be used to consume the trained model.

The embedded features of the XGBoost are used in the proposed flow diagram. As a result, the feature engineering step is fed by the feed from feature importance. Label encoding also occurred. We proposed label encoding, which converts categorical features into numbers. The list of categorical features includes deployment environment encoded to (1 for cloud, 2 for on-premises), operating system environment encoded to (1 for Linux, 2 for Windows), data storage type encoded to (2 for SSD type), data transformation engine encoded to (1 for Hadoop, 2

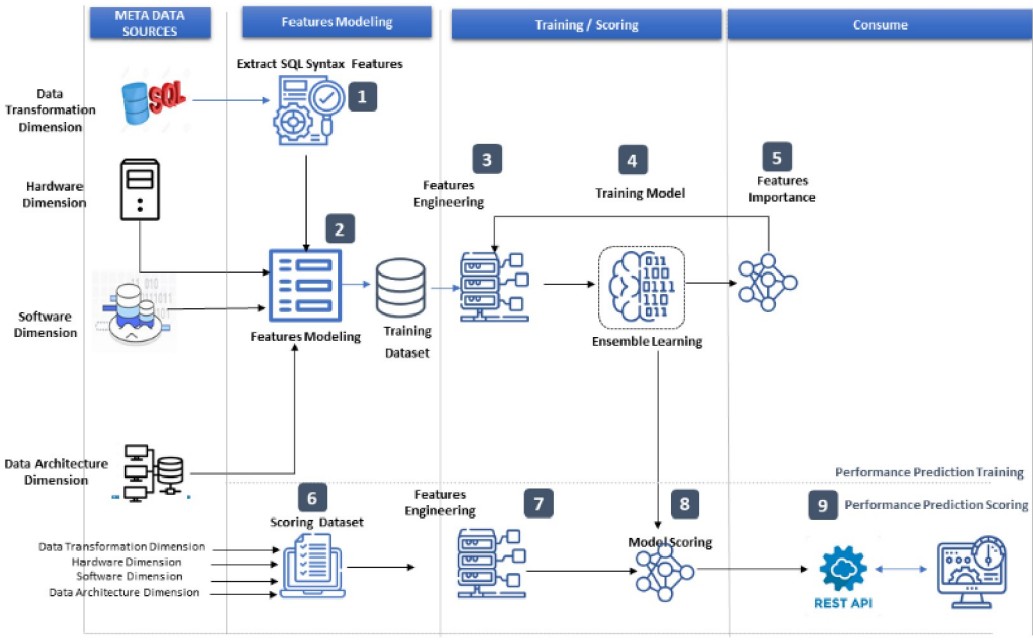

**Fig 1. Proposed framework.**

for Oracle, and 3 for Greenplum), and query number encoded to 1 through 10 because we have 10 queries for data transformation.

## Training dataset metrics

Here, an example of the features extracted is presented. From query syntax statement features, hardware features, software features, and data architecture features to historical performance logs, these metrics will be used as an input to train the machine learning model. However, this list of metrics is just an example of what we can feed the model; as already demonstrated, the data preparation phase will take place to prepare training datasets. So, the objective of this study is to show how to extract features from hardware, software, SQL syntax, and data architecture. It does not show the complete list of features, which will change from case to case, as shown in Table 1. However, the majority of queries are complex, with sub queries and complex joins, as shown in the attached S1 File, which was used in the experiment work. The concept was to use complex queries from real-world implementations [the reference project was for a telecom operator on a big data platform (cloudera environment) that handles streaming data]. These features will be derived from query syntax statement specifications, hardware specifications, software specifications, data architecture specifications, and historical performance logs. Feature modeling is used to extract input features from various types of queries.

## Ensemble learning XGBoost algorithm

Gradient boosting is the original XGBoost model that combines weak basic learning models with stronger learning in an iterative [35]. At each iteration of the gradient boost, the residual

**Table 1. Training dataset metrics.**

| Dimension | Features |
|---|---|
| Hardware | ■ Number of used processors |
| | ■ Env_Deploymnet_Type |
| | ■ Env_OS_Type |
| Software | ■ ETL_Engine_Type |
| | ■ Target_DataBase_Engine_Type |
| | ■ Hadoop_Target_DataStore_Type |
| | ■ Data_Virtualization_Engine |
| SQL | ■ Data_Transformation_No_Conditions |
| | ■ Data_Transformation_No_String_Conditions |
| | ■ Data_Transformation_No_Encoding_String_Conditions |
| | ■ Data_Transformation_No_Sub_Query |
| | ■ Data_Transformation_No_Agg |
| | ■ Data _Transformation_Frequency_Rate |
| | ■ Data _Transformation _No_Format_Func |
| Data Architecture | ■ DataSources_Type (Structure File, Structure DB, Structure Stream data, unstructured File, unstructured stream data, etc) |
| | ■ Target_DataBase_Engine_Type (RDBMS, Columnar, In-Memory) |
| | ■ Data_Transformation_Engine_Type (ETL_Engine_File, ETL_Engine_Repos, ETL_Engine_Repos_Pushdown) |
| | ■ Number_Data_Transformation_sources |
| | ■ Number_Data_Transformation_Files_sources |
| | ■ Number_Data_Transformation_Hadoop_sources |
| | ■ Hybrid_Data_Sources (Files with DB, DB with Hadoop, files with Hadoop) |
| | ■ BigData_Data_Transformation_Engine_Type |
| | ■ Data_Architrecture_Type (lambda, Kappa) |

will be used to correct the previous predictor so that the specified loss function can be optimized. As an improvement, regularization is added to the loss function to determine the objective function of the XGBoost model performance measurement. So, we can write the model as,

$$\mathcal{Y}_i = \sum_{k=1}^{K} f_k(x_i), \; f_k \in \mathcal{F} \tag{5}$$

Where K is the number of trees, is a function in the functional space and is the set of all possible classification and regression trees (CARTs). The objective function always needs to contain training loss and regularization [35]. Xgboost takes a more iterative approach to determining the best K. The equation shows the iteration numbers that should be used in the XGBoost to achieve the smallest possible error. The number of rounds parameter is used as an input parameter in the XGBoost. The ensemble technique is based on a cleverer approach because it is based on many iterations that are combined to perform the final one. Rather than training all the models separately, boosting trains models that succeed, with each new model being trained to correct the mistakes made by the previous ones. Models are added in a sequential order until no further advancements can be made. So, rather than specifying an exact number, we provide the algorithm with a set of parameters to determine the optimal number of trees (no of estimators).

$$obj(\theta) = \sum_{i}^{n} l(y_i, \mathcal{Y}_i) + \sum_{k=1}^{K} \Omega f_k \tag{6}$$

we need to learn are those functions $f_i$, each containing the structure of the tree and the leaf scores. Learning tree structure is much harder than traditional optimization problem where we can simply take the gradient. It is intractable to learn all the trees at once. Instead, we use an additive strategy: fix what we have learned and add one new tree at a time. We can write the prediction value at step t as $\mathcal{Y}_i^{(t)}$ then the objective equation will be [36].

$$obj^{(t)} = \sum_{i=1}^{n} l(y_i, \mathcal{Y}_i^{(t-1)}) + f_t(x_i)] + \Omega(f_t) + C \tag{7}$$

If we consider using mean squared error (MSE) as our loss function, the objective becomes [36].

$$obj^{(t)} = \sum_{i=1}^{n} \left[ l(\mathcal{Y}_i^{(t-1)} - y_i) + g_i f_t(x_i) + \frac{1}{2} h_i f^2_t(x_i) \right] + \Omega(f_t) + C \tag{8}$$

where the $g_i$ and $h_i$ are defined as,

$$g_i = \partial_{\mathcal{Y}_i^{(t-1)}} l(y_i, \mathcal{Y}_i^{(t-1)}) \tag{9}$$

$$h_i = \partial^2_{\mathcal{Y}_i^{(t-1)}} l(y_i, \mathcal{Y}_i^{(t-1)}) \tag{10}$$

After we remove all the constants, the specific objective at step t becomes the optimization goal for the new tree. We can optimize every loss function as follows [36]:

$$\sum_{i=1}^{n} [g_i f_t(x) + \frac{1}{2} h_i f^2_t(x_i)] + \Omega(f_t) \tag{11}$$

We have introduced the training step, but we still need to consider the regularization term. We need to define the complexity of the tree. Here ω is the vector of scores on leaves, and $T$ is

the number of leaves. So, we define the complexity as [36]:

$$\Omega(f_t) = \gamma T + \frac{1}{2}\lambda \sum_{j=1}^{T} \omega_j^2 \tag{12}$$

So, the best objective reduction we can get in the equation, $\omega_j$ are independent with respect to each other, the form $G_i\omega_j(x_i) + \frac{1}{2}\left(H_j + \lambda\right)\omega^2_j(x_i)$ is quadratic and using the best $\omega_j$, we can compute the optimal weight $\omega_j^*$ of leaf j as [36]:

$$\omega_j^* = -\frac{G_i}{H_j + \lambda} \tag{13}$$

$$obj^* = -\frac{1}{2}\sum_{j=1}^{T}\frac{G_j^2}{H_j + \lambda} + \gamma T \tag{14}$$

Then, it can be used as a scoring function to measure the quality of the tree structure q. This score is like the impurity score for evaluating decision trees, except that it is derived from a wider range of objective function [36].

$$Gain = \frac{1}{2}\left[\frac{G_L^2}{H_L + \lambda} + \frac{G_R^2}{H_R + \lambda} + \frac{G_R^2 + G_L^2}{H_R + H_L + \lambda}\right] - \gamma \tag{15}$$

## Experimental work description

The experimental work will use performance logs for ten SQL queries that have already been deployed in different environments. These performance logs, along with environmental requirements and query syntax, will be used to prepare the training dataset features as outlined in the next section, which will explain how to prepare the training dataset. What is the specification of the environment used to extract these training datasets? What are the specifications of the experimental environment? What are the results of the proposed framework? An example of the queries is shown in Table 2.

### Environment specification

The environmental specifications used in the experiment work are shown in Table 3.

### Training dataset

The total dataset records used in the experiment was around 27225 records with 45 variables. The training data set had around 24502 records (90% of the total dataset) and cross-validation of around 2723 records (10% of the whole dataset). The number of selected variables is 40. The following variables have been ignored: Start Time, End Time, Day, Size DT DB Sources, DT Records Number. The dataset is based on the performance tracking logs for ten queries that ran in different environments, as is illustrated in the environment specification section. The query syntax metrics are used (data source type, number of data sources, number of conditions in the query syntax, number of string conditions in the query syntax, number of subqueries in the query syntax, number of aggregated functions in the SQL statement, number of data sources in the query syntax, number of formatting functions in the query syntax, and so on). The hardware metrics (number of processors, ram used, hard disk type (HDD or SSD)), environment deployment (on-cloud, on-premises), number of nodes in the Hadoop environment, number of edge nodes, edge node processors, edge node storage, number of worker nodes, worker node processors, worker node storage, number of master nodes, master node

**Table 2. Example of query syntax used in the experiment work.**

| No | Syntax | Freq |
|---|---|---|
| **Q1** | insert into table sandbox_nxn.hive_tcpdr partition (h) | H |
| | select s.last_timestamp_utc,cast(trim(s.session_duration) as bigint), cast(s.last_timestamp_utc as bigint), s.roaming, cast(s.msisdn as bigint), cast(s.imei as bigint), cast(s.tac as bigint), cast(s.last_location_type as bigint), cast(s.last_cell as bigint), cast(s.volume_ul as bigint), cast(s.volume_dl as bigint), s.id_content_provider, s.id_content_category, s.first_lac_enodeb_id, s.first_location_id, s.last_lac_enodeb_id, s.last_location_id, year(from_unixtime(cast(s.last_timestamp_utc as bigint))), month(from_unixtime(cast(s.last_timestamp_utc as bigint))),day(from_unixtime(cast(s.last_timestamp_utc as bigint))), hour(from_unixtime(cast(s.last_timestamp_utc as bigint))) as h | |
| | from | |
| | sandbox_nxn.${var0} as s where s.last_timestamp_utc is not null and cast (s.msisdn as bigint) is not null; | |
| **Q2** | INSERT into sandbox_nxn. fact_pos_mbb_hourly partition (day) | D |
| | SELECT MBB.LAST_TIMESTAMP_HOUR, MBB.duration_seconds, MBB.number_of_visits, MBB.MSISDN, MBB.ROAMING_FLAG, MBB.VOLUME_UL, MBB.VOLUME_DL, MBB.AVG_THROUGHPUT, POS_CUST.subscriber_id, CPRV.DIM_CONTENT_PROVIDER_DK, DT.timekey, DD.dw_date_calendar_dim_key, case WHEN CELL.dw_dim_cell_site_key IS not NULL THEN CELL.dw_dim_cell_site_key else CELL2.dw_dim_cell_site_key end, case WHEN CELL.dw_dim_cell_site_key IS not NULL THEN CELL.dw_dim_site_location_key else CELL2.dw_dim_site_location_key end, COALESCE(HNDST.dw_dim_handset_key,181653), NULL, CASE WHEN MBB.last_cell_location_type = 0 then '2G' WHEN MBB.last_cell_location_type = 1 then '3G' WHEN MBB.last_cell_location_type in(3,4) then '4G' ELSE 'NA' END, MBB.day as day | |
| | FROM sandbox_nxn.mbb_tcpdr_hourly as MBB | |
| | Join | |
| | sandbox_nxn.dim_pos_customer_uniq as POS_CUST on (MBB.MSISDN = cast(concat('965',POS_CUST.SUBSCRIPTION_NO) as bigint)) LEFT OUTER join offload.dim_date_calendar as DD on (MBB.DAY = concat(cast(DD.dw_month_num_of_year as string),concat(concat('-',cast(DD.dw_day_num_of_month as string)),concat('-',DD.dw_year_id)))) Left OUTER join sandbox_nxn.dim_cell_site_uniq as CELL on (lpad(MBB.last_location_id,4,'0') = lpad(CELL.CELLHEX,4,'0') and CELL.end_date is null and (last_cell_location_type in (3,4) and lpad(MBB.last_lac_enodeb_id,5,'0') = CELL.SITEHEX)) left outer join sandbox_nxn.dim_cell_site2_uniq as CELL2 on (lpad(MBB.last_location_id,4,'0') = lpad(CELL2.CELLHEX,4,'0') and CELL2.end_date is null and (last_cell_location_type in (0,1) and lpad(MBB.last_lac_enodeb_id,5,'0') = CELL2.lachex)) join sandbox_nxn.dim_content_provider as CPRV on (substr(MBB.ID_CONTENT_PROVIDER,2,length(MBB.ID_CONTENT_PROVIDER)-2) = CPRV.ID_CONTENT_PROVIDER) join sandbox_nxn.dim_time as DT on (MBB.LAST_TIMESTAMP_HOUR = DT.HOUR) LEFT OUTER JOIN offload.dim_handset_imei as HNDST ON (cast(MBB.TAC as string) = HNDST.TAC) WHERE MBB.MSISDN IS NOT NULL and MBB.LAST_TIMESTAMP_HOUR = ${var0} and MBB.day = '${var1}' and POS_CUST.TERMINATION_DATE IS NULL; | |

processors, mast node processors, etc. are used. Metrics of data architecture such as database engine, Hadoop engine, and Hadoop hybrid database were used. The used list of features is shown in Table 4. The list of features in Table 4 is not limited to these features, it is part of the proposed framework that can be added more.

**Table 3. Environment specification used in the experiment work.**

| HW Name | HW Specs |
|---|---|
| **Performance Log Environment** | Processors = 16 |
| | RAM = 128 |
| | HD SSD = 20 TB |
| | Database Engine = Greenplum |
| **Performance Log Environment** | Hadoop Environment |
| | 6 Cluster Nodes |
| | Each Node |
| | Processors = 16 |
| | RAM = 64 |
| | HD SSD = 10 TB |
| | (One Mater Node, One Edge Node, 4 Worker Nodes) |
| | Database Engine = Hive /Impala |
| **Experiment Environment** | Processors = 16 |
| | RAM = 64 |
| | Database Engine = Oracle HD SSD = 20 TB |

**Table 4. List of features in the training dataset.**

| Features Domain | Features |
|---|---|
| **Hardware** | Environment |
| | Processors |
| | RAM |
| | Env_Deploymnet |
| | Env_OS |
| | Data_Storage_Type |
| | Number_of_nodes_for_Hadoop_Environment |
| | Number_of_Edge_Nodes |
| | Edge_Processors |
| | Edge_RAM |
| | Edge_Storage |
| | Number_of_Workers_Nodes |
| | Workers_Processors |
| | Workers_RAM |
| | Worker_Storage |
| | Number_of_Master_Nodes |
| | Master_Processors |
| | Master_RAM |
| | Master_Storage |
| **Software** | Data_Transformation_Engine_Type |
| **Data Architecture** | Number_ Data_Transformation_sources |
| | Number_ Data_Transformation_Files_sources |
| | Number_ Data_Transformation_DB_sources |
| | Number_ Data_Transformation_Hadoop_sources |
| | Size_ Data_Transformation_Files_Sources |
| | Size_ Data_Transformation_DB_Sources |
| | Size_ Data_Transformation_Hadoop_sources |
| **SQL syntax** | Data_Transformation_No_Conditions |
| | Data_Transformation_No_String_Conditions |
| | Data_Transformation_No_En_String_Conditions |
| | Data_Transformation_No_Sub_Query |
| | Data_Transformation_No_Agg |
| | Data_Transformation_No_Format_Func |
| | Data_Transformation_Frequency_Rate |
| | DT_Records_Number |
| **Performance** | Hour_of_the_data |
| | Query |
| | Start_time |
| | End_time |
| | Day |
| | Performance |

## Features engineering

The process of encoding the training dataset was used for data preparation. We have coded the numbers for the categorical features. The list of categorical features includes deployment environment encoded to (1 for cloud, 2 for on-premises), operating_system_environment encoded to (1 for Linux, 2 for Windows), data_storage_type encoded to (2 for SSD type),

data_transformation_engine encoded to (1 for Hadoop, 2 for Oracle, and 3 for Greenplum), and query number encoded to 1 till 10 as we have 10 queries as data transformation syntax. We dropped Start Time, End Time, and Day from the feature selection because they had already been used to calculate the performance (target feature) (Size of DT DB Sources, DT Records Number) to be redundant (DT Data Source Size, Number of Records); we reformatted the day features to be used (YYYYMMDD).

### Development steps

The experimental life cycle is illustrated as features modeling, features engineering, model development, then testing and validation. The feature modeling extracts the features from the query syntax statement characteristics, hardware characteristics, data processing engine characteristics, data architecture characteristics, and historical performance logs using different environments. Feature engineering checks the correlation and importance across all input variables and target variables. Then machine learning development, model development and model validation.

### XGBoost alogrithm parameters

This section describes the algorithm parameters shown in Table 5.

### XGBoost alogrithm flow

This section describes the algorithm flow steps that are shown in Table 6.

**Table 5. Algorithm parameters.**

| Parameter | Parameter Value | Parameter Description |
|---|---|---|
| booster | gblinear | Select the type of model to run at each iteration. It has two options: |
| | | gbtree: tree-based models |
| | | gblinear: linear models |
| eta | 0.0004 | The learning rate used to weight each model |
| max_depth | 10 | The maximum depth of each tree |
| gamma | 10 | Gamma is a pseudo-regularization parameter |
| subsample | 0.9 | Represents the fraction of observations to be sampled for each tree |
| colsample_bytree | 1 | Number of features (columns) used in each tree |
| objective | reg: squarederror | reg: squarederror: for linear regression |
| eval_metric | rmse | rmse –root mean square error. |
| nrounds | 30000 | max number of boosting iterations. |
| nthreads | 10 | Number of threads can also be manually specified via |
| early_stopping_rounds | 20 | If NULL, the early stopping function is not triggered. If set to an integer k, training with a validation set will stop if the performance doesn't improve |
| verbose | **0** | If 0, xgboost will stay silent. If 1, it will print information about performance. If 2, some additional information will be printed out |
| | Results | The model performance measures the tested dataset that compared the predicted values versus actual values, the tested dataset being around 2723 records. |
| | | The mean absolute error (MAE) is given by. |
| | | $MAE = 1/N \sum_{i=1}^{n} |y_i - \widehat{y_i}| = 0.663$ |
| | | The root mean square error (RMSE) is given by. |
| | | $RMSE = \sqrt{1/N \sum_{i=1}^{n} (y_i - \widehat{y_i})^2} = 0.814$ |
| | | The average percentage difference between the actual and the predicted performance is equal to 5.3%, as shown in Fig 4. |

**Table 6. Algorithm flow.**

| |
|---|
| 1. Import Libraries |
| 2. Loading Data |
| 3. Reformat Loaded Data |
| 4. Split the Data (Training /Testing) |
| 5. Convert the cleaned data frame. |
| 6. XGBoost parameters Configuration |
| 7. Model Training |
| 8. Extract the results. |
| 9. Compare observed vs predicted. |
| 10. Extract Feature's importance |
| 11. Check Model Performance |

## Results

The results show the importance of embedded features in XGBoost. Analyzing the results in Fig 2, we observe the environment OS, data storage type, number of data sources (Number of DT sources), number of SQL sub queries (Number of DT Sub Query) in the SQL query, etc.

The training model behavior on a tested dataset includes the predicted performance values from the model versus the actual observed performance, as shown in Fig 3.

The model performance measures the tested dataset that compared the predicted values versus actual values, the tested dataset being around 2723 records.

The mean absolute error (MAE) is given by.

$$MAE = 1/N\sum_{i=1}^{n}|y_i - \widehat{y_i}| = 0.663 \tag{16}$$

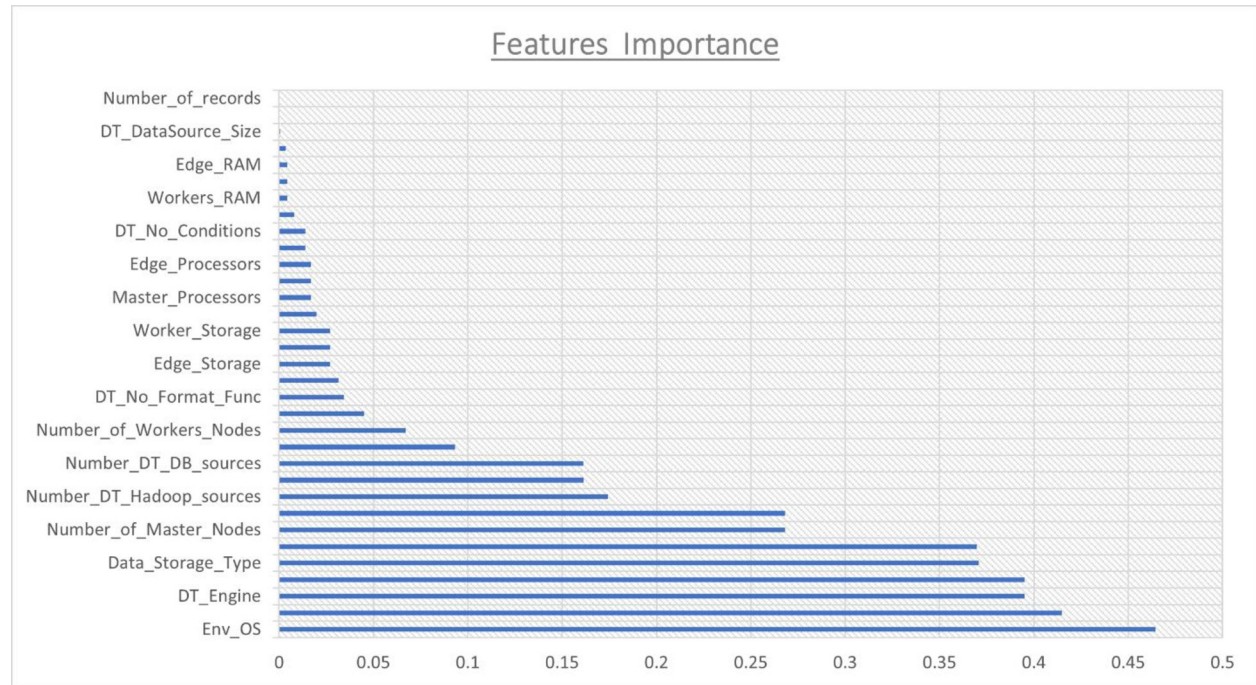

**Fig 2. Feature's importance results.**

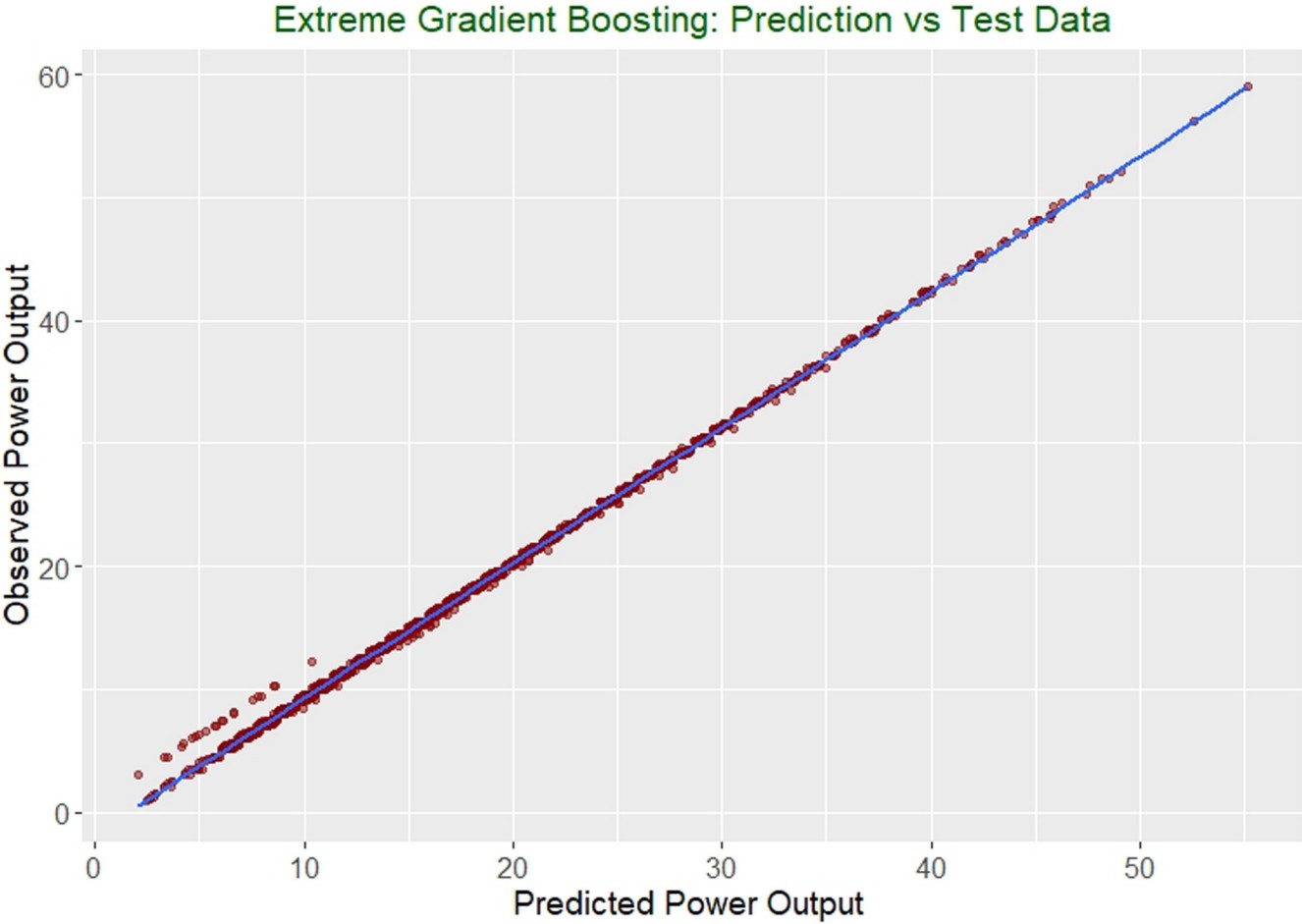

**Fig 3. Model behavior predicted versus actual tested dataset.**

The root mean square error (RMSE) is given by.

$$RMSE = \sqrt{1/N \sum_{i=1}^{n} (y_i - \widehat{y_i})^2} = 0.8143 \qquad (17)$$

The average percentage difference between the actual and the predicted performance is equal to 5.3%, as shown in Fig 4.

## Discussion

The key lessons of learning from the proposed approach and other approaches in the related work section are shown in Table 7. Number of features. Most of the approaches in the related work section limit the number of features used to train the prediction model to improve model accuracy. While the proposed approach did not limit the features, it did propose a generalization approach for features modelling to enrich the prediction model's input training data set. The impact of feature standardization is that it standardizes the features used for training the prediction model despite differences in the query's environment specifications in terms of hardware and software, as well as data architecture. So, defining a feature domain, then extracting a list of features from each domain, generalizes an approach to standardizing

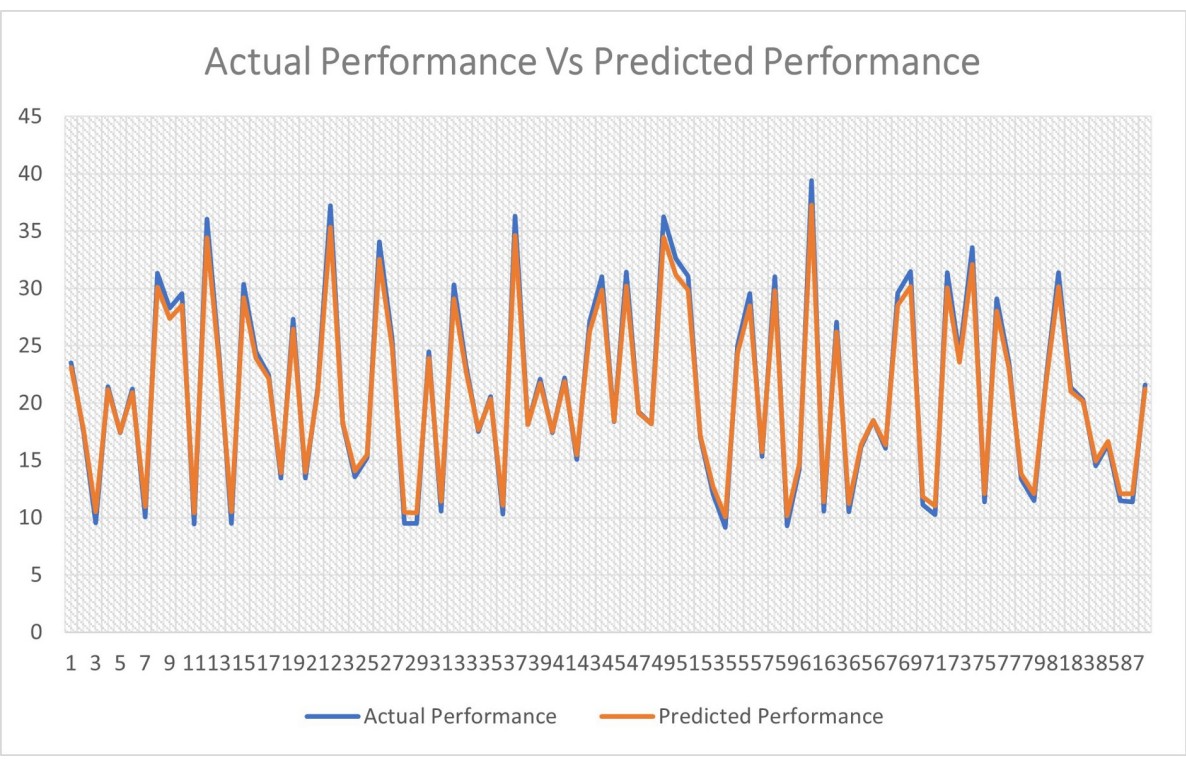

**Fig 4. Actual performance vs predicted.**

the features. This step of standardizing the input features for the query prediction model is not considered in any of the related works. All the related work depends on the optimizer statistics to extract the logs and the features that impacted the query performance as per the historical execution plans. While the proposed approach standardized the input features considering the performance logs. Features Modelling is a step towards modelling the features, not to limiting the features. Other approaches just extract limited features. It is clear from the related work using the ensemble learning technique that the boosting methodology has shown better performance in such regression problems. A comparison between the related approach and the proposed approach is shown in Table 7.

## Conclusion

This paper proposed a framework for predicting query performance. The challenges of developing such a predictive performance are how to standardize the required features to train the model, which machine learning technique to be used, so the trained model can be used as an optimization tool or simulation tool to optimize the structure query, hardware specification, software specification, or data architecture specification. Thus, for these domains (software, hardware, data architecture, SQL syntax), a list of features extracted from these domains were proposed by the framework, as shown in sections 4, 5 and 10. Then, the framework was used to develop a performance prediction model with an ensemble learning technique based on extreme boosting methodology, considering the challenges of using linear regression. The ensemble learning technique (tree-based model) handles missing values because it has the built-in ability to handle missing values. Dealing with linear and non-linear regression issues, handling over-fitting with built-in regularization of L1 (lasso regression), and L2 (ridge

**Table 7. Comparison between proposed approach and related works.**

| Points | Classifier Approach | BOW Approach | KCCA Approach | Proposed Approach | Notes |
|---|---|---|---|---|---|
| Number of Features | Limited Features | Limited Features | Limited Features | No Limited Features | |
| Features Standardization | NA | NA | NA | Features Modeling | |
| Features Importance | NA | grid search | NA | Embedded in the ensemble learning | |
| Machine Algorithm Approach | Classifier with any prediction algorithm | SVR, RFR and GBR | kernel canonical correlation analysis | XGBoost | |
| Software Features Optimization | NA | NA | NA | Supported | |
| Hardware Features Optimization | NA | NA | NA | Supported | |
| Query Syntax Optimization | NA | NA | NA | Supported | |
| Data Architecture Features Optimization | NA | NA | NA | Supported | |
| approach performance | MAE = NA  RMSE = NA  Actual vs predicted = 0.077 | MAE = 0.8518  RMSE = NA | MAE = NA  RMSE = NA  Actual vs predicted = 0.45 | MAE = 0.663  RMSE = 0.8143  Actual vs predicted = 0.053 | Some relative approaches measures MAE, actual vs predicted. However, to predict the response time for a query, most query performance approaches rely on DBMS optimizing statistics and the cost estimation of each operator in the query execution plan, which also focuses on resource utilization (CPU, I/O). Modeling query features is thus a critical step in developing a robust query performance prediction model. In this paper, we propose a new framework based on query feature modeling and ensemble learning to predict query performance and use this framework as a query performance predictor simulator to optimize the query features that influence query performance. |

regression), which prevented over-fitting of the model. That is why the regularized form of the GBM (gradient boosting machine) is also called the XGBoost. The proposed model performance showed an average difference of around 5.3% between the actual and the predicted performance.

There are areas of enhancement that may be considered in the future in the proposed framework. Instead of relying on the feature's importance embedded in ensemble learning, we can use separate feature selection steps to be added to the development cycle of machine learning. The second area is that we can use ensemble deep learning to develop a performance prediction model instead of using the ensemble learning technique. The automation of the feature modelling process improves the process of data preparation. One of the automation mechanisms is to use NLP to extract the SQL features instead of manually analyzing them. The proposed framework can be enriched by all these future contributions.

## Supporting information

**S1 File.**
(RAR)

## Author Contributions

**Conceptualization:** Mohamed Zaghloul.

**Data curation:** Mohamed Zaghloul.

**Methodology:** Mohamed Zaghloul, Amr Ali-Eldin.

**Resources:** Mohamed Zaghloul.

**Supervision:** Mofreh Salem, Amr Ali-Eldin.

**Validation:** Mofreh Salem, Amr Ali-Eldin.

**Writing – original draft:** Mohamed Zaghloul.

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
