## [Decision Letter · Decision Letter 0]

3 Jun 2021

PONE-D-21-16156

A New Framework based on Features Modeling and Ensemble Learning to Predict Query Performance

PLOS ONE

Dear Dr. Zaghloul,

Thank you for submitting your manuscript to PLOS ONE. After careful consideration, we feel that it has merit but does not fully meet PLOS ONE’s publication criteria as it currently stands. Therefore, we invite you to submit a revised version of the manuscript that addresses the points raised during the review process.

Based on the comments received from the reviewers and my own observation, I recommend minor revisions for the article.

We look forward to receiving your revised manuscript.

Kind regards,

Thippa Reddy Gadekallu

Academic Editor

PLOS ONE

Journal Requirements:

"Mohamed Mustafa Zaghloul"

6. Thank you for stating the following in your Competing Interests section: 

"On behalf of all authors, disclose any competing interests that could be perceived to bias this work—acknowledging all financial support and any other relevant financial or non-financial competing interests."

Reviewers' comments:

Reviewer's Responses to Questions

**Comments to the Author**

1. Is the manuscript technically sound, and do the data support the conclusions?

Reviewer #1: Yes

Reviewer #2: Yes

2. Has the statistical analysis been performed appropriately and rigorously? 

Reviewer #1: Yes

Reviewer #2: Yes

3. Have the authors made all data underlying the findings in their manuscript fully available?

Reviewer #1: Yes

Reviewer #2: Yes

4. Is the manuscript presented in an intelligible fashion and written in standard English?

Reviewer #1: Yes

Reviewer #2: Yes

5. Review Comments to the Author

Reviewer #1: 1. Did the authors concentrate on optimization on joins and other complex queries such as nested sub queries in their experiment?

2. It will be better for the readers if the author presents the parameters of the proposed NN model in a tabular column (epochs, architecture infos and so on).

3. What is the optimal value of 'k' in equation 7 that the authors have considered?

4. In the section "FEATURES ENGINEERING", the authors have discussed few pre-processing stages such as categorical data resolving and feature removal. Does the authors consider any feature selection algorithms such as filter methods, wrapper methods, for picking/dropping features?

5. It is advised to compare the author's proposed work with any other standard work especially with the works discussed in section two (Related Work).

Please cite the following works to enhance the feature engineering-related works.

1. Ashokkumar P, Siva Shankar G, Gautam Srivastava, Praveen Kumar Reddy Maddikunta, and Thippa Reddy Gadekallu. 2021. A Two-stage Text Feature Selection Algorithm for Improving Text Classification. ACM Trans. Asian Low-Resour. Lang. Inf. Process. 20, 3, Article 49 (April 2021), 19 pages. DOI:https://doi.org/10.1145/3425781

2. G. Siva Shankar, P. Ashokkumar, R. Vinayakumar, Uttam Ghosh, Wathiq Mansoor, Waleed S. Alnumay, "An Embedded-Based Weighted Feature Selection Algorithm for Classifying Web Document", Wireless Communications and Mobile Computing, vol. 2020, Article ID 8879054, 10 pages, 2020. https://doi.org/10.1155/2020/8879054

Reviewer #2: • The authors should emphasize the difference between other methods to clarify the position of this work further.

• The Wide ranges of applications need to be addressed in Introductions

• The objective of the research should be clearly defined in the last paragraph of the introduction section.

• Add the advantages of the proposed system in one quoted line for justifying the proposed approach in the Introduction section.

• The motivation for the present research would be clearer, by providing a more direct link between the importance of choosing your own method.

In feature extraction and ensemble learning the authors can refer An ensemble based machine learning model for diabetic retinopathy classification. An ensemble machine learning approach through effective feature extraction to classify fake news. For prediction the authors can refer Predictive model for battery life in IoT networks

6. PLOS authors have the option to publish the peer review history of their article (what does this mean?). If published, this will include your full peer review and any attached files.

Reviewer #1: No

Reviewer #2: No

---

## [Author Response · Author response to Decision Letter 0]

9 Aug 2021

Dear Editor, 

I would like to thank you and the reviewers for the fruitful comments which have helped us improve our paper. Kindly find attached the following files:

- Response on Reviewer comments 

- Manuscript with the updated marked section 

- Final updated Manuscript

We have done our best to meet reviewers’ comments and hope that the paper is suitable for publication.

“The authors received no specific funding for this work”.

Many thanks 

Sincerely, 

Mohamed Mustafa 

Corresponding Author

---

## [Editor Report · Decision Letter 1]

28 Sep 2021

A New Framework based on Features Modeling and Ensemble Learning to Predict Query Performance

PONE-D-21-16156R1

Dear Dr. Zaghloul,

We’re pleased to inform you that your manuscript has been judged scientifically suitable for publication and will be formally accepted for publication once it meets all outstanding technical requirements.

Kind regards,

Usman Qamar

Academic Editor

PLOS ONE
---

## [Editor Report · Acceptance letter]

30 Sep 2021

PONE-D-21-16156R1 

A New Framework based on Features Modeling and Ensemble Learning to Predict Query Performance 

Dear Dr. Zaghloul:

I'm pleased to inform you that your manuscript has been deemed suitable for publication in PLOS ONE. Congratulations! Your manuscript is now with our production department. 

Kind regards, 

on behalf of

Dr. Usman Qamar 

Academic Editor

PLOS ONE